# WHEN A ROBOT IS MORE CAPABLE THAN A HUMAN: LEARNING FROM CONSTRAINED DEMONSTRATORS

**Xinhu Li**[1]  **Ayush Jain**[1,2†]  **Zhaojing Yang**[1,3]  **Yigit Korkmaz**[1]  **Erdem Bıyık**[1]
[1]Thomas Lord Department of Computer Science, University of Southern California
[2]Meta AI  [3]Department of Computer Science & Engineering, University of California San Diego
`lixinhu98@gmail.com, ayushj240@meta.com, zhy070@ucsd.edu,`
`{ykorkmaz, biyik}@usc.edu`

## ABSTRACT

Learning from demonstrations enables experts to teach robots complex tasks using interfaces such as kinesthetic teaching, joystick control, and sim-to-real transfer. However, these interfaces often constrain the expert's ability to demonstrate optimal behavior due to indirect control, setup restrictions, and hardware safety. For example, a joystick can move a robotic arm only in a 2D plane, even though the robot operates in a higher-dimensional space. As a result, the demonstrations collected by constrained experts lead to suboptimal performance of the learned policies. This raises a key question: *Can a robot learn a better policy than the one demonstrated by a constrained expert?* We address this by allowing the agent to go beyond direct imitation of expert actions and explore shorter and more efficient trajectories. We use the demonstrations to infer a state-only reward signal that measures task progress, and self-label reward for unknown states using temporal interpolation. Our approach outperforms common imitation learning in both sample efficiency and task completion time. On a real WidowX robotic arm, it completes the task in 12 seconds, 10× faster than behavioral cloning. We provide real-robot videos and additional resources on our project website: https://sites.google.com/view/constrainedexpert.

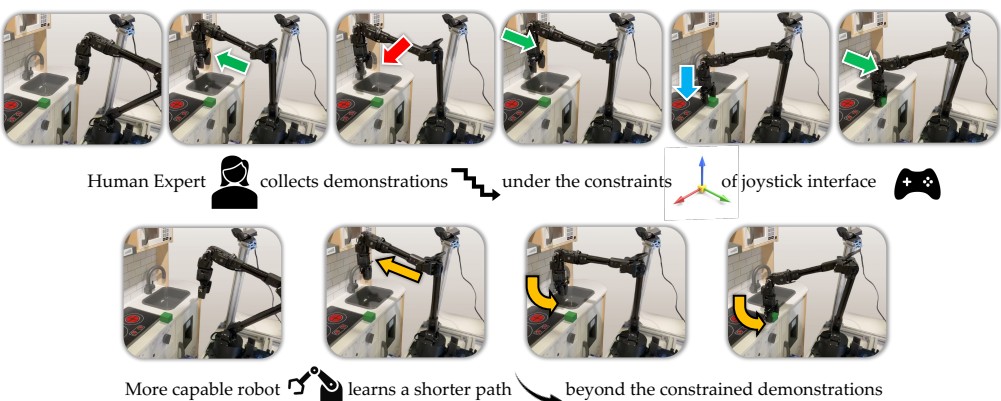

Figure 1: A human expert constrained by a mode-switching joystick produces segmented trajectories. A robot employing LfCD-GRIP executes smooth and efficient motions beyond the demonstrations.

# 1 INTRODUCTION

Imitation learning (IL) and inverse reinforcement learning (IRL) are powerful frameworks to acquire complex robotic behaviors from expert demonstrations (Argall et al., 2009; Abbeel et al., 2010; Arora & Doshi, 2021; Hussein et al., 2017; Ravichandar et al., 2020; Bıyık et al., 2022). However, in practice, human operators are often constrained by the control interface, occluded viewpoints, or physical precision, preventing them from demonstrating optimal behaviors. For example, a 6-DoF

---

[†]Ayush Jain performed this work in his personal capacity outside of his work at Meta AI.

arm teleoperated with a joystick interface (Herlant et al., 2016; Losey, 2020) requires mode-switching to control different axes. This restricts their ability to perform smooth, simultaneous, and multi-axis motions. As a result, the demonstrations exhibit slow, segmented demonstration trajectories (Figure 1). In contrast, the robot is capable of fast, fluid, and coordinated actions across all degrees of freedom.

This discrepancy raises the key problem of **Learning from Constrained Demonstrations (LfCD)**: Can an agent learn from constrained demonstrations and discover more efficient behaviors once those constraints are lifted? Traditional approaches struggle because IL (Schaal, 1999) mimics the suboptimal actions, while IRL (Ho & Ermon, 2016) infers reward functions that reflect the same constraints. While work on learning from suboptimal demonstrations (Gao et al., 2018; Brown et al., 2019) addresses imperfect or noisy expert behavior, LfCD applies to competent experts whose demonstrations are constrained by the interface, leading to goal-directed but inefficient trajectories.

The LfCD problem poses three key challenges to enable agents to explore beyond constrained expert imitation. (1) Since expert actions are restricted by the interface, the IRL reward should be decoupled from the expert action, defined for state-state transitions rather than state-action. (2) Since demonstrations cover only part of the state space, a learning agent must identify which explored states have reliable reward estimates. (3) Even for the novel states encountered during exploration, the agent requires a generalizable reward signal.

To address these challenges, we introduce **LfCD** with **G**oal-proximity **R**eward **InterP**olation (LfCD-GRIP). (1) To decouple rewards from constrained expert actions, our key insight is to use a state-only measure of progress toward the goal. We adopt a **goal proximity** reward (Lee et al., 2021; Bae et al., 2025), trained along expert demonstration trajectories via backward temporal decay from the goal. However, such estimates do not generalize to observations beyond the demonstration distribution. (2) To ensure reliable reinforcement learning, LfCD-GRIP includes a **confidence estimator** that identifies expert-like observations where goal proximity reward is valid. (3) Finally, to assign rewards in novel states never visited by the experts, it **interpolates** proximity values along agent-collected trajectories, propagating task progress smoothly between the states covered in demonstrations. *Intuitively, we use expert-like observations as anchors and interpolate proximity values across the agent's own trajectories to propagate the notion of task progress beyond the constraints.*

We evaluate LfCD-GRIP across a range of discrete and continuous control tasks in both navigation and manipulation domains. Empirical results show that LfCD-GRIP consistently outperforms baseline IL and IRL approaches by finding efficient task solutions, particularly in scenarios where demonstrations are constrained. For instance, in a real-world pick-and-place task using a WidowX arm, LfCD-GRIP reduces task completion time from 100 seconds (under IL) to just 12 seconds. This demonstrates the effectiveness of our method in enabling robots to go beyond the constraints of the demonstrations.

**Our contributions in this work are as follows:**

1. We introduce the problem of learning from constrained demonstrations (LfCD) to highlight that expert demonstrations are often constrained in practice, limiting the quality of learned behaviors.
2. We propose LfCD-GRIP, an IRL framework that extends goal-proximity rewards beyond expert demonstrations with a confidence-based proximity interpolator over an agent's online learning.
3. We show LfCD-GRIP enables more efficient and generalizable policies across multiple domains, outperforming existing IL and IRL baselines under various constrained expert demonstrators.

## 2    RELATED WORK

**Imitation Learning and Learning from Observations.** Imitation learning (IL) enables robots to acquire behaviors by mimicking expert demonstrations without requiring access to an explicit reward function (Argall et al., 2009; Hussein et al., 2017; Schaal, 1999). The most basic form of IL is behavioral cloning (BC) (Pomerleau, 1988), which treats imitation as a supervised learning problem by directly mapping states to actions. Generative adversarial imitation learning (GAIL) (Ho & Ermon, 2016) introduced an adversarial training framework that matches the state-action distribution of expert demonstrations. These methods replicate the expert's actions, which is suboptimal in LfCD.

Imitation learning from observations addresses the case when expert action labels are unavailable (Liu et al., 2018; Torabi et al., 2018a; Yang et al., 2019; Liu et al., 2020; Ko et al., 2024; Wen et al., 2024). For instance, GAIL from Observations (GAIfO) (Torabi et al., 2018b) extends GAIL to learn solely from state transitions. It mitigates action space mismatch but still replicates expert-like

state transitions. In contrast, LfCD-GRIP only follows the expert's demonstrated intended goal and discovers improved policies that leverage the robot's full capabilities.

**Learning from Suboptimal Demonstrations.** Expert demonstrations can be noisy or suboptimal due to limited skill or inconsistent behavior (Choi et al., 2019; Yang et al., 2022; Zhu et al., 2022; Yu et al., 2023; Gao et al., 2018; Zhang et al., 2021). T-REX (Brown et al., 2019) and D-REX (Brown et al., 2020) infer reward functions from suboptimal demonstrations by ranking trajectory segments, while Self-Supervised Reward Regression (SSRR) (Chen et al., 2021) learns a reward by injecting noise into expert trajectories and using noise levels as self-supervised ranking signals. In contrast, LfCD assumes the demonstrator arrives at the correct goal with actions optimal *within* their constrained action space, but is limited by real-world factors such as interface restrictions or safety constraints. Moreover, our method explicitly leverages the goal-directed nature of tasks to enable policy improvement beyond the constraints, which is not leveraged by prior works on learning from suboptimal demonstrations.

**Learning from Cross-Embodied Demonstrators.** Recent approaches, such as those by Raychaudhuri et al. (2021); Hu et al. (2022); Dasari et al. (2020); Zakka et al. (2022); Xu et al. (2023), learn from experts operating in different action spaces than the robot, due to mismatched embodiment or control interfaces. They typically learn alignment functions or explicit mappings that translate expert demonstration actions into actions feasible within the robot's action space (Liang et al., 2025). For example, Cross-Domain Imitation Learning (CDIL) (Raychaudhuri et al., 2021) employs optimal transport to align action distributions before imitation. Similarly, robot-aware control (Hu et al., 2022) models the dynamics of both the expert and the robot to bridge the action space mismatch. These techniques aim to reproduce the expert's behavior, whereas we aim to discover *better* policies that exploit the agent's wider action space.

**Goal-Proximity Reward Learning from Demonstrations.** Proximity-based IRL (Lee et al., 2021; Ma et al., 2023) trains a goal proximity function from demonstrations to provide shaped, dense, action-free rewards that measure task progress. However, it fails to generalize to states observed in the agent's online exploration beyond the demonstration distribution. As a result, the agent receives low reward in unexplored states, limiting its ability to discover more efficient policies than those demonstrated. More recent approaches like ReWiND (Zhang et al., 2025) and Robometer (Liang et al., 2026) try to address this issue by leveraging large robotics datasets and/or large pretrained vision-language models. LfCD-GRIP, on the other hand, takes an orthogonal approach: it addresses the issue by interpolating proximity values along agent trajectories in online rollouts. Using high-confidence states as anchors, we construct smoother, more reliable rewards that guide exploration beyond constrained demonstrations.

## 3 LfCD Problem Formulation

We formulate the LfCD problem as a Markov decision process (MDP) (Sutton, 1984), defined by the tuple $\langle \mathcal{S}, \mathcal{A}, R, P, \rho_0, \gamma \rangle$, where $\mathcal{S}$ is the state space, $\mathcal{A}$ is the action space, $R : \mathcal{S} \times \mathcal{A} \times \mathcal{S} \to \mathbb{R}$ is the reward function, $P(s' \mid s, a)$ is the transition distribution, $\rho_0$ is the initial state distribution, and $\gamma \in [0, 1)$ is the discount factor. A policy $\pi(a \mid s)$ defines a distribution over actions conditioned on the current state. The objective is to find a policy that maximizes the expected discounted return,

$$\max_\pi \mathbb{E}_{(s_0, a_0, \ldots, s_T) \sim \pi} \left[ \sum_{t=0}^{T-1} \gamma^t R(s_t, a_t, s_{t+1}) \right] \tag{1}$$

where $T$ is the episode length. Our setting focuses on goal-reaching problems, where the primary objective is to complete a task by arriving at a desired terminal state. A key structural property of such tasks is that progress toward the goal can be measured in a state-dependent manner, independent of the specific actions taken. This structural assumption enables us to decouple reward learning from expert actions under constraints. Without access to the reward function $R$, this objective is achieved by learning from a dataset of expert demonstrations $\mathcal{D}^e = \{\tau_1, \tau_2, \ldots, \tau_K\}$, where $\tau_k = \{s_0, a_0, \ldots, s_T\}$. We work in a generalized formulation where access to actions is not necessary and state trajectories $\tau_k = \{s_0, s_1, \ldots, s_T\}$ are sufficient to define the objective.

Particularly in LfCD, expert demonstrations are collected under *action space constraints* that limit the expert's available actions at each state. We denote this *potentially unknown* constrained action space as $\mathcal{A}^e(s) \subseteq \mathcal{A}$, indicating that at state $s$, the expert can only choose from actions $a \in \mathcal{A}^e(s)$. In contrast,

the learning agent i.e., the robot, has access to the full action space $\mathcal{A}$, and can potentially learn policies that outperform the constrained expert by utilizing actions unavailable during demonstration[*].

## 4    APPROACH: LFCD WITH GOAL-PROXIMITY REWARD INTERPOLATION

To address the problem of learning from constrained demonstrations, we develop the **G**oal-proximity **R**eward **I**nter**P**olation (LfCD-GRIP) framework, which extends proximity-based IRL with confidence-guided reward propagation. Our approach builds on the insight that expert actions are restricted by the interface, but their demonstrations still contain reliable signals of task progress. LfCD-GRIP therefore (i) defines a goal-proximity reward that depends only on states, decoupling reward from suboptimal expert actions, (ii) introduces a confidence estimation module to identify which states—whether from expert data or agent rollouts—provide trustworthy proximity values, and (iii) incorporates a trajectory-wise interpolation mechanism that propagates the reliable proximity values to novel states encountered during exploration. Together, these components enable the agent to explore efficient behaviors toward the goal and surpass the constraints of human demonstrators (Figure 1).

### 4.1    GOAL-PROXIMITY AS ACTION-FREE REWARD

Proximity-based IRL (Lee et al., 2021) defines rewards based on the estimated proximity of a state to the task goal, rather than relying on expert actions. This formulation assumes that expert demonstrations are optimal and collected without constraints. The proximity function $f_\phi(s)$ is trained with two complementary objectives: (1) expert states are assigned exponentially decayed proximity values, such that states closer to the goal receive higher values, and (2) agent rollouts are pushed toward zero to avoid overgeneralization. The combined loss is

$$\mathcal{L}_\phi = \underbrace{\mathbb{E}_{s_t \sim \mathcal{D}^e} \left( f_\phi(s_t) - \delta^{T-t} \right)^2}_{\mathcal{L}_\phi^e} + \underbrace{\mathbb{E}_{s_t \sim \mathcal{D}^r} \left( f_\phi(s_t) \right)^2}_{\mathcal{L}_\phi^r} \tag{2}$$

where $\delta \in (0, 1)$ is the temporal decay factor, $T$ is the trajectory length, $\mathcal{D}^e$ the expert dataset, and $\mathcal{D}^r$ the dataset of agent rollouts. We denote $\mathcal{L}_\phi^e$ as the expert supervision loss, and $\mathcal{L}_\phi^r$ as online regularization loss, for reference in later sections.

Proximity-based IRL alternates between updating the proximity network and training the policy with rewards derived from it. The reward label for a state transition is the reduction in goal proximity:

$$\hat{R}_{\text{prox}}(s_t, s_{t+1}) = f_\phi(s_{t+1}) - f_\phi(s_t) \tag{3}$$

The policy $\pi_\theta$ is trained to maximize the expected cumulative reward:

$$\max_\theta \; \mathbb{E}_{\pi_\theta} \left[ \sum_t \hat{R}_{\text{prox}}(s_t, s_{t+1}) \right] = \max_\theta \; \mathbb{E}_{\pi_\theta} \left[ \sum_t f_\phi(s_{t+1}) - f_\phi(s_t) \right] \tag{4}$$

While our framework is compatible with any reinforcement learning algorithm, we use proximal policy optimization (PPO) (Schulman et al., 2017) for all experiments. Proximity-based IRL provides a dense, progress-based reward signal independent of expert actions, making it well-suited for learning from demonstrations collected under constrained action spaces.

**Limitation of proximity-based IRL**: All agent explored states are assigned low proximity, de-incentivizing exploration. While effective in unconstrained settings, Proximity-based IRL struggles when expert demonstrations are collected under constrained action spaces. This limitation stems from the objective for online states, $\mathcal{L}_\phi^r$, which penalizes proximity predictions on all states outside the demonstration distribution—including those that could enable shorter paths to the goal. As a result, it discourages exploration and prevents the agent from discovering more efficient solutions.

To overcome this, we propose to provide meaningful proximity values for out-of-distribution observations with expert demonstrations. The key idea is to propagate proximity estimates from reliable observations—those with confident and well-calibrated predictions—to uncertain online observations. As training progresses and more reliable observations are identified, this propagation extends to a broader region of the observation space. This process introduces two main challenges: (1) identifying which observations have reliable proximity predictions, and (2) assigning proximity values to

---

[*]Our method trivially extends to the cases where the robot's action space is also state-dependent as long as $\mathcal{A}^e(s) \subseteq \mathcal{A}^r(s)$ for all $s \in \mathcal{S}$.

---

**Algorithm 1** LfCD-GRIP

---

**Require:** Expert dataset $\mathcal{D}^e$, decay factor $\delta$, rollout budget $N$, RL algorithm (e.g., PPO)
 1: Initialize proximity network $f_\phi$, policy $\pi_\theta$
 2: Pretrain $f_\phi$ using expert loss $\mathcal{L}_\phi^e$
 3: **for** iteration $= 1$ to $N$ **do**
 4:     Collect rollouts $\mathcal{D}^r$ using policy $\pi_\theta$
 5:     **Proximity Model Training:**
 6:     Estimate confidence for each $s_t \in \mathcal{D}^r$ using MCD
 7:     Identify confident endpoints and construct sub-trajectories $\mathcal{D}^{\mathrm{conf}}$
 8:     Generate interpolated proximity targets for intermediate states from $\mathcal{D}^{\mathrm{conf}}$
 9:     Compute proximity loss: $\mathcal{L}_\phi^{\mathrm{GRIP}} = \mathcal{L}_\phi^e + \mathcal{L}_\phi^{\mathrm{conf}} + \mathcal{L}_\phi^{\mathrm{unconf}}$
10:     Update $f_\phi$ using gradient descent on $\mathcal{L}_\phi^{\mathrm{GRIP}}$
11:     **Policy Training:**
12:     Compute rewards for policy buffer with $\hat{R}_{\mathrm{prox}}(s_t, s_{t+1}) = f_\phi(s_{t+1}) - f_\phi(s_t)$
13:     Update policy $\pi_\theta$ via RL using proximity rewards
14: **end for**

---

uncertain observations based on their relationship to confident anchors. We address both through two core components: a confidence estimation module and a trajectory-wise interpolation mechanism.

## 4.2 GOAL-PROXIMITY CONFIDENCE ESTIMATION MODULE

This module aims to identify reliable observations that can serve as anchors for proximity propagation. We first treat expert observations as reliable, as their proximity values are predefined based on temporal distance to the goal. However, we must also distinguish reliable observations among online collected samples, which lie outside the expert distribution.

Monte Carlo Dropout (MCD) (Gal & Ghahramani, 2016) provides a practical solution. By enabling dropout at inference time and performing multiple stochastic forward passes, we can estimate the uncertainty of the proximity predictions. The variance of these predictions reflects model uncertainty, with lower variance indicating higher confidence:

$$\mathrm{confidence}_\phi(s_t) = -\mathrm{Var}(f_\phi(s_t)) = -\frac{1}{K}\sum_{k=1}^{K}\left(f_\phi^{(k)}(s_t) - \bar{f}_\phi(s_t)\right)^2,$$

where $f_\phi^{(k)}(s_t)$ denotes the output of the proximity network on the $k$-th forward pass with dropout, and $\bar{f}_\phi(s_t) = \frac{1}{K}\sum_{k=1}^{K} f_\phi^{(k)}(s_t)$.

In practice, we pretrain the proximity network on expert demonstrations using the expert supervision loss $\mathcal{L}_\phi^e$, which results in low-variance (i.e., high-confidence) predictions on those expert observations. To classify whether an online-collected state is high-confidence or not, we compute a dynamic threshold at each iteration based on the expert states. Specifically, the confidence threshold is set as the maximum proximity variance observed among expert states. Any online state with lower variance than this threshold is treated as high-confidence. This design guarantees that all expert states are always included as high-confidence anchors throughout training, while also allowing online states with similarly low uncertainty to be used for proximity interpolation.

## 4.3 GOAL-PROXIMITY INTERPOLATION MECHANISM

Once high-confidence observations are identified, we propagate their proximity values to nearby low-confidence observations. Here, *nearby* refers to temporal rather than spatial proximity. To enable this propagation, we identify sub-trajectories where both endpoints are high-confidence, and use them as anchors for interpolation. Concretely, when two high-confidence observations lie on the same trajectory, the intermediate states are assigned smoothly interpolated proximity values, as illustrated in Figure 2. For each intermediate state $s_t$, we define a proximity target $\hat{f}_t$ by linearly interpolating in the log-proximity space

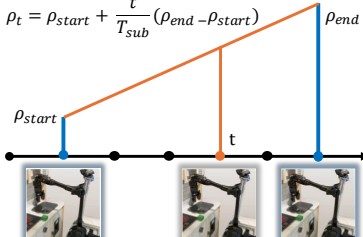

Figure 2: Proximity is interpolated between high-confidence anchors.

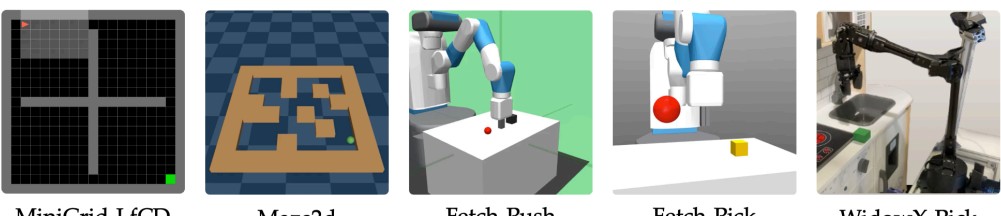

| MiniGrid-LfCD | Maze2d | Fetch-Push | Fetch-Pick | WidowX-Pick |

Figure 3: We use various manipulation and navigation tasks with different kinds and degrees of constrained expert demonstration datasets.

between the sub-trajectory's start and end points:

$$\hat{f}_t = \delta^{\rho_{\text{start}} + \frac{t}{T_{\text{sub}}}(\rho_{\text{end}} - \rho_{\text{start}})} \tag{5}$$

where $T_{\text{sub}}$ is the temporal length of the sub-trajectory, and $\rho_{\text{start}}$ and $\rho_{\text{end}}$ are the log-scale goal-proximity distances at its endpoints.

To stabilize training in the early stages, we introduce an annealing strategy that gradually increases the reliance on interpolated values. At each training iteration, we sample a masking probability $p_{itr} \in [0,1]$ that starts at 1 and linearly decays to 0 over time. With probability $p$, we mask the interpolated targets, assigning them zero proximity. This encourages conservative learning early on, and enables progressive generalization as training proceeds. The resulting loss for interpolated states becomes:

$$\mathcal{L}_\phi^{\text{conf}} = \mathbb{E}_{s_t \sim \mathcal{D}^{\text{conf}}} \left[ (1 - m_{itr}) \cdot \left( f_\phi(s_t) - \hat{f}_t \right)^2 + m_{itr} \cdot (f_\phi(s_t))^2 \right] \tag{6}$$

where $m_{itr} \sim \text{Bernoulli}(p_{itr})$ is a stochastic mask applied independently to each intermediate state, and $\mathcal{D}^{\text{conf}}$ denotes online sub-trajectories with high-confidence start and end states. This masking strategy allows the model to interpolate proximity values only when it becomes confident enough, ensuring smooth propagation without introducing premature bias from uncertain data. We include a comparison to the no-masking variant in Appendix C.

For all remaining states—those not covered by confident sub-trajectories—we retain the original assumption of zero proximity:

$$\mathcal{L}_\phi^{\text{unconf}} = \mathbb{E}_{s_t \sim (\mathcal{D}^r \setminus \mathcal{D}^{\text{conf}})} (f_\phi(s_t))^2 \tag{7}$$

The full training objective for the proximity function in LfCD-GRIP is:

$$\mathcal{L}_\phi^{\text{GRIP}} = \mathcal{L}_\phi^e + \mathcal{L}_\phi^{\text{conf}} + \mathcal{L}_\phi^{\text{unconf}} \tag{8}$$

The complete LfCD-GRIP training loop, including proximity updates and policy optimization, is provided in Algorithm 1. In summary, LfCD-GRIP trains an agent with PPO by inferring a goal-proximity reward complemented with confidence estimation and proximity interpolation from constrained expert demonstrations.

## 5 EXPERIMENTS

We investigate the effectiveness of LfCD-GRIP through the following experiments: (1) Can LfCD-GRIP learn to produce optimal trajectories that other methods fail to discover? (2) Does LfCD-GRIP outperform standard IL methods and state-of-the-art approaches for learning from suboptimal demonstrations under constrained experts? (3) Can LfCD-GRIP lead to policies that leverage actions unavailable to the expert? (4) How do LfCD-GRIP and baselines perform as expert constraints become more severe? (5) What is the practical impact of LfCD-GRIP for real-robot applications?

**Baselines and Ablations**. We compare LfCD-GRIP against common imitation learning and inverse RL baselines, a state-of-the-art method for learning from suboptimal demonstrations, and ablations of our approach to validate the technical contributions. Our primary evaluation metric is the average trajectory length. When computing the average trajectory length, we include both successful and unsuccessful trajectories. Unsuccessful episodes are not terminated early; instead, their trajectory length is set to the maximum episode horizon.

- **BC** directly maps observations to actions via supervised learning on expert demonstrations.

- **GAIL** trains a discriminator to distinguish expert observation-action pairs from those generated by the learning agent, using the discriminator as a reward function.

- **GAIfO** removes the need for expert actions by matching state-transition distributions.

- **SSRR** learns a reward by ranking demonstrations with injected noise and using noise severity as a proxy for suboptimality. It is a state-of-the-art method for learning from suboptimal demonstrations.

- **Proximity** (Proximity-based IRL) learns a reward function based on the temporal distance to the goal and trains the agent via RL.

- **Proximity-Drop** is an ablation of Proximity-based IRL with dropout layers enabled, but without confidence estimation or interpolation. This baseline isolates the contribution of our proposed modules from the regularizing effect of dropout.

- **LfCD-GRIP** augments Proximity-Drop with confidence estimation and interpolation to propagate reliable proximity values to unseen observations.

## 5.1 LFCD-GRIP DISCOVERS SHORTCUT TRAJECTORIES TO GOAL IN MINIGRID-LFCD

To evaluate whether LfCD-GRIP can recover optimal trajectories beyond those demonstrated, we design a simple but illustrative MiniGrid environment (Brockman et al., 2016; Chevalier-Boisvert et al., 2023). The agent always starts in the top-left corner with the goal fixed in the bottom-right. Expert demonstrations, constrained to the four cardinal directions, traverse only the top row and rightmost column. In contrast, the agent is allowed to move in all eight directions, including diagonals. This asymmetry introduces a shorter diagonal path that lies well outside the expert distribu-

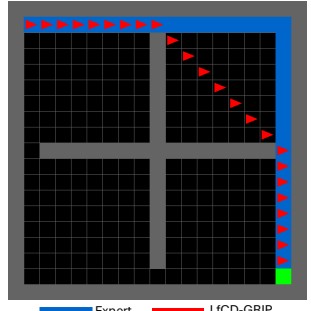 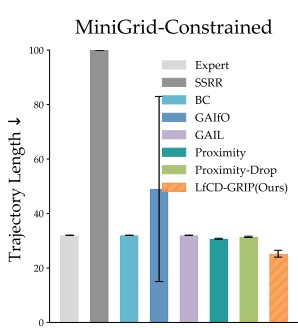

Figure 4: MiniGrid-LfCD Results. (left) The expert follows the blue path to the green goal, while LfCD-GRIP takes the red shortcut; (right) average episode length across methods.

tion. As shown in Figure 4, only LfCD-GRIP discovers this optimal shortcut by propagating goal proximity values to unseen but reachable states, while all baselines remain confined to the demonstrated path, yielding longer average trajectories.

## 5.2 QUANTITATIVE RESULTS ON CONSTRAINED EXPERT DEMONSTRATIONS

We further validate LfCD-GRIP on various navigation and manipulation tasks, as shown in Figure 3: MAZE2D (Fu et al., 2020; Kumar et al., 2020), FETCHPICK (Plappert et al., 2018), FETCHPUSH (Plappert et al., 2018). Details for all environments are provided in Appendix A.

**Action Space Constraints.** To simulate limited control interfaces, we constrain the expert's action space during demonstration collection. The full action space for continuous environments is normalized to $[-1, 1]$ in each dimension. The specific constraints for each environment are as follows:

- MAZE2D. The robot uses 2D accelerations with experts actions constrained to $[-0.1, 0.1]$.
- FETCHPICK. The robot actions control the 3D continuous Cartesian displacements $(x, y, z)$ and a binary action for the gripper. Expert actions are constrained to $[-0.1, 0.1]$.
- FETCHPUSH. The action space matches that of FetchPick except the gripper, which is disabled. Expert actions are constrained to $[-0.05, 0.05]$.

This setup emulates realistic scenarios in which robots are capable of high-speed motion, but expert demonstrations are collected under constrained control for safety and reliability.

For each environment, we compare all methods under two settings: (1) the *UnconstrainedExpert* setting, where the agent and the expert share the same constrained action space, except in Maze2D , where both the agent and expert use the full action space. This intentional exception allows us to

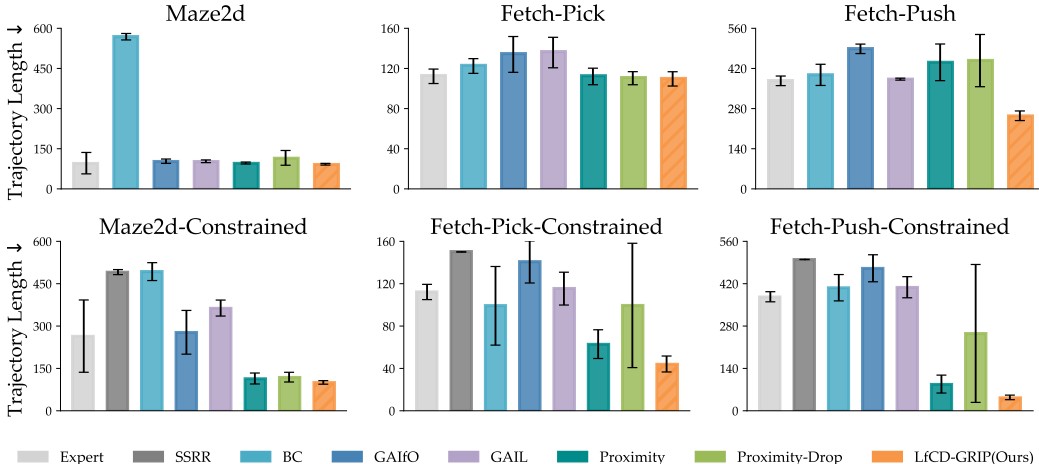

Figure 5: Average episode length across *UnconstrainedExpert* settings (top) and *ConstrainedExpert* settings (bottom). LfCD-GRIP consistently outperforms all baselines in constrained settings by finding short trajectory length solutions consistently, and remains robust in unconstrained ones.

contrast Maze2D with Minigrid; and (2) the *ConstrainedExpert* setting, where the agent has access to the full action space while the expert demonstrations are generated under constraints. This dual setup assesses LfCD-GRIP's performance gains when constraints are introduced in expert demonstrations.

**Results Discussion**. For all environments, we run experiments with four random seeds, and each evaluation checkpoint averages results over 160 episodes. We report the average episode length across all evaluation trajectories from the final trained policy, including unsuccessful attempts. This length metric measures the policy's optimality and ability to leverage the unconstrained action space for faster goal completion. In Figure 5, LfCD-GRIP achieves strong performance across all environments in both settings. Although it remains competitive in *UnconstrainedExpert*, it consistently outperforms other baselines when the agent operates with an expanded action space in *ConstrainedExpert*. In the Maze2D-Constrained setting, although the bar for Proximity-based IRL appears visually similar to LfCD-GRIP in the figure, LfCD-GRIP completes the task in 100 transitions on average—reducing episode length by over 10% compared to Proximity-based IRL, which requires 113 transitions.

These results support our central claim: LfCD-GRIP enables agents to go beyond expert constraints by learning a goal-proximity reward function, rather than mimicking constrained expert behavior.

### 5.3 ANALYSIS: DOES LfCD-GRIP LEVERAGE OUT-OF-CONSTRAINT (OOC) ACTIONS?

To assess whether LfCD-GRIP generalizes beyond constrained demonstrations, we analyze the proportion of actions selected by each agent that fall outside the expert's action space in the Maze2D environment. Table 1 reports both the success rate and the ratio of out-of-constraint (OOC) actions. LfCD-GRIP achieves a 100% success rate while selecting OOC actions 100% of the time. In contrast, GAIL and BC favor

| Baseline | Success Rate | OOC Action Ratio |
|---|---|---|
| GAIL | 69% | 71% |
| BC | 12% | 69% |
| GAIfO | 51% | 100% |
| **LfCD-GRIP** | **100%** | **100%** |

Table 1: Success rate and OOC action ratio in Maze2D-Constrained. LfCD-GRIP achieves 100% success while effectively leveraging OOC actions.

in-distribution actions, and GAIfO, despite using OOC actions entirely, fails to achieve high task success. These results underscore the importance of reward generalization, not just action diversity. This analysis further validates that our method is action-independent, as it successfully exploits actions beyond expert constraints to achieve optimal performance.

### 5.4 ANALYSIS: LfCD-GRIP PERFORMANCE WITH MORE SEVERE EXPERT CONSTRAINTS

We evaluate LfCD-GRIP under two constraint levels in the FetchPush environment. In the relaxed case, the constraint is widened to $[-0.7, 0.7]$, allowing more expressive expert behavior. In the severe case (Severity 2), the expert's action space is limited to $[-0.05, 0.05]$, simulating highly restricted demonstrations. We compare against representative subset of baselines: BC (supervised imitation),

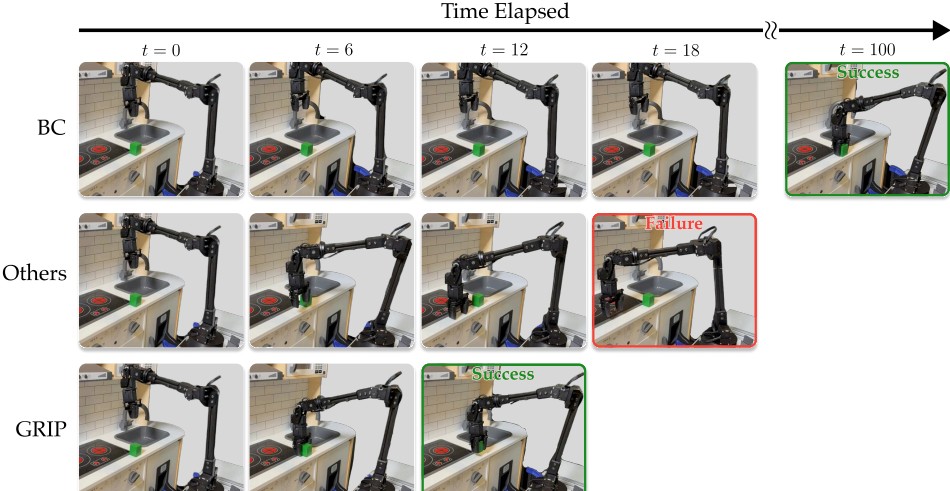

Figure 7: Real-robot rollouts of the WidowX-Pick task. Only BC learns meaningful policies, while LfCD-GRIP completes the task **10x faster** than BC.

Expert (expert performance), and Proximity-based IRL (closest to our method). We omit GAIL and GAIfO, as they never surpass expert performance and behave similarly to the Expert baseline. SSRR is excluded because it fails to learn a successful policy under constrained demonstrations.

Figure 6 shows LfCD-GRIP maintains strong performance across both constraint levels, whereas baselines such as BC and Proximity-based IRL degrade substantially under severe constraints. These results demonstrate that LfCD-GRIP works effectively across varying degrees of expert action space constraints, and is able to find short path solutions to the goal consistently by utilizing the agent's exploration efficiently.

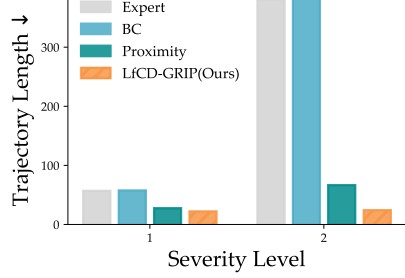

## 5.5 WIDOWX-PICK:
### SIMULATION AND REAL-ROBOT EXPERIMENT

We evaluate LfCD-GRIP on the WidowX-Pick task, both in simulation and on the real WidowX 250s robotic arm. We use a mode-switching joystick interface (Losey, 2020)

Figure 6: Varying constraint severity shows the increasing benefit of LfCD-GRIP over baselines. Severity 2 means constraint $[-0.05, 0.05]$.

to collect demonstrations, which allows control of only one axis at a time. This creates a natural constraint in the expert's action space, yielding constrained demonstrations. Training in simulation (Figure 8) shows that LfCD-GRIP outperforms all baselines, achieving substantially shorter trajectories. Except for BC, the other baselines fail to learn meaningful policies, while BC remains limited by the expert's constrained behavior.

We then deploy the learned policy on the real-world WidowX-Pick setup (Figure 7). While BC reproduces expert-like behavior, it fails to utilize the full action space and executes slowly, requiring 100 seconds per trial. In contrast, LfCD-GRIP generates efficient trajectories that completes the task **10x faster**, in just 12 seconds. These results demonstrate that our method transfers effectively to real hardware and enables better-than-expert performance by overcoming action space constraints.

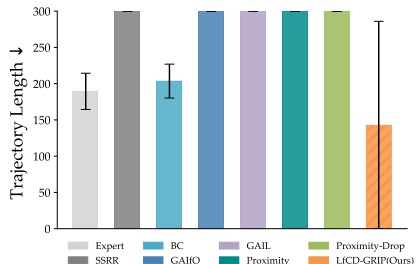

## 6 CONCLUSION & LIMITATIONS

We address the challenge of learning from constrained expert demonstrations, where the expert lacks access to

Figure 8: WidowX-Pick Simulation. Only BC and LfCD-GRIP succeed, with LfCD-GRIP being more efficient.

the robot's full action space due to interface or embodiment limitations. We propose LfCD-GRIP,

a framework that learns a generalizable progress-aware reward function via interpolation, enabling agents to extrapolate beyond constrained demonstrated behavior and discover more efficient policies. Extensive experiments show that LfCD-GRIP outperforms baselines, highlighting the promise of decoupling reward learning from constrained expert actions.

**Limitations.** The proximity-based reward assumes that task progress can be measured with respect to a specific goal state. While well suited for goal-reaching tasks, this limits applicability to settings without clearly defined terminal conditions. Extending LfCD-GRIP to such tasks remains an important direction for future work. Estimating progress in multi-task scenarios also remains challenging, as our approach currently relies on goal-conditioned proximity estimates tailored to individual demonstrations. Generalizing progress signals across tasks with semantically varied goals will require advances in representation learning and reward modeling.

**Future Work.** While we demonstrate LfCD-GRIP in the setting of constrained experts, the core idea of interpolating proximity-based rewards along an agent's own trajectories is broadly applicable. In particular, this mechanism enables reward generalization to unseen states whenever the expert data provides only partial coverage of the state space, which in turn enables more efficient paths to the goals than those observed in demonstrations. We therefore view constrained experts as one practically important instance of a more general problem: *learning progress-aware rewards from sparse or biased expert coverage*, and plan to investigate our approach more broadly.

## REPRODUCIBILITY STATEMENT

To ensure reproducibility, We provide the source code in the supplementary material, with a README file containing commands for all experiments. We describe the simulation environments in Appendix A, the real-robot setup in Appendix B, and the training details in Appendix E and Appendix F.

## LLM USAGE

We used large language models (LLMs) to assist with grammar correction and rewording. No model-generated content was used for scientific claims, experiments, or core contributions. All ideas and analyses are original and developed by the authors.

## ACKNOWLEDGEMENTS

We acknowledge funding by the Airbus Institute for Engineering Research (AIER) for this project. We thank the USC Center for Advanced Research Computing (CARC) for providing us with compute resources. We thank Ziyi Liu for valuable discussions and for helping improve the clarity and presentation of the paper.

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

# Appendix

## A  ENVIRONMENT DETAILS

We conduct experiments across four environments: MINIGRID-LFCD, MAZE2D, FETCHPICK, and FETCHPUSH. Below we describe their environment dynamics, state/action spaces, and demonstration collection protocols. Note that the full action space for continuous environments is always normalized to $[-1, 1]$ in each dimension.

**Minigrid-LfCD.** This environment is a grid-based navigation task with discrete spatial observations. Each cell in the grid is encoded as a one-hot vector representing one of four categories: wall, empty space, the agent, or the goal. The layout of the grid remains fixed throughout all episodes. The environment is fully observable, with observations of size $[19 \times 19 \times 4]$.

The agent always starts in the top-left corner, and the goal is located in the bottom-right corner. The full action space $A$ includes 8 discrete movement actions: the four cardinal directions (up, down, left, right) and four diagonals. In contrast, the expert is constrained to only the 4 cardinal directions, simulating limited action capabilities.

The expert dataset consists of a single demonstration, generated using a breadth-first search (BFS) planner that computes the shortest path under the constrained action space. This setup ensures that the demonstration is optimal given the expert's limitations, while allowing the agent to potentially discover shorter paths using the full action space.

**Maze2D.** The agent is a point mass that navigates through a 2D maze by controlling its $(x, y)$ acceleration in continuous space. The state includes the agent's 2D position, velocity, and the goal position. We slightly modify the standard `maze2d-medium-v1` environment from D4RL (Fu et al., 2020) by reducing the maximum episode length from 600 to 400 steps.

The full action space $A$ is a 2-dimensional continuous space, where each dimension controls acceleration in the $x$ or $y$ direction. In the constrained setting, the expert's actions are clipped to a restricted range $[-0.1, 0.1]$, reducing movement magnitude and limiting directional flexibility.

For training, we collect two datasets: 800 expert demonstrations using the full action space, and 800 expert demonstrations under the $[-0.1, 0.1]$ constrained action space, both using the planner provided by D4RL.

**FetchPick and FetchPush.** These manipulation tasks are adapted from the OpenAI Gym Fetch environments (Plappert et al., 2018), where a 7-DoF arm controls its end-effector in 3D space with an additional continuous dimension for the gripper (which is ineffective in FETCHPUSH). The 16-dimensional state vector includes the relative position of the goal to the object, the end-effector to the object, and the robot's joint configuration. Following prior findings from Proximity-based IRL (Lee et al., 2021), we exclude velocity information from the state input, which improves performance for learning-from-observation approaches.

For both environments, the full action space $A$ is a 4-dimensional continuous space, representing Cartesian displacements in $x$, $y$, and $z$ directions of the end effector, along with a gripper control signal, which is fixed for FetchPush. In the constrained setting, the expert is limited to actions within a smaller bounded region, $[-0.1, 0.1]$ for FETCHPICK and $[-0.05, 0.05]$ for FETCHPUSH , reducing dexterity and making successful grasps more challenging.

For both FETCHPICK and FETCHPUSH, we collect 400 constrained demonstrations using a scripted policy that moves the gripper above the object, descends to grasp/push it, and transports it to the goal.

## B  WIDOWX DESCRIPTION

**Environment.** We use the ManiSkill simulator to collect 550 expert demonstrations and pretrain the policy, followed by sim-to-real transfer to the WidowX 250s hardware. The task requires the robot to grasp a cube placed on a surface and lift it slightly above that surface to succeed. To simplify the task and reduce orientation complexity, the robot's end-effector is fixed in a downward-pointing orientation. Additionally, to mitigate challenges in precisely replicating visual setups between simulation and the real-world hardware, we use a low-dimensional observation space instead of visual inputs. Specifically, the observations provided to the robot include the gripper's end-effector position,

the gripper opening state, the cube's position and whether the cube is grasped. To ensure clarity of the cube's state for the agent, the cube is initialized randomly at one of three fixed, predetermined positions at each environment reset.

**Expert Action Space.** Expert demonstrations are collected in simulation using a Machenike G5 controller. To minimize accidental inputs and ensure precise control, we map discrete movements to the controller's directional pad (D-pad) and ABXY buttons. Specifically, the D-pad is used to command horizontal movements—left, right, forward, and backward—while the A and Y buttons control vertical movement (up and down). The B and X buttons control the opening and closing of the gripper. Human demonstrators thus issue discrete, single-axis commands sequentially, restricting simultaneous multi-axis control and limiting the range and complexity of demonstrated actions.

**Agent Action Space.** The robot agent operates in a continuous 4-dimensional action space: three degrees for Cartesian movement and one for gripper actuation. Unlike the human expert, the agent can perform smooth and simultaneous multi-axis movements, enabling more efficient trajectories and improved manipulation behaviors.

## C  FURTHER ABLATION: NO MASKING

To evaluate the importance of the masking strategy in LfCD-GRIP, we conduct an ablation study in which the masking probability is removed—i.e., the interpolated values are always used as training targets for intermediate states. This variant is evaluated on the FetchPick-Constrained environment.

As shown in Figure 9, removing the masking leads the agent to become overconfident in its early interpolations. This results in reward propagation through unreliable states, ultimately preventing the policy from generalizing and achieving successful task completion. These findings highlight the importance of gradual interpolation: masking helps regulate learning by limiting reward propagation to only confident regions in the early stages of training.

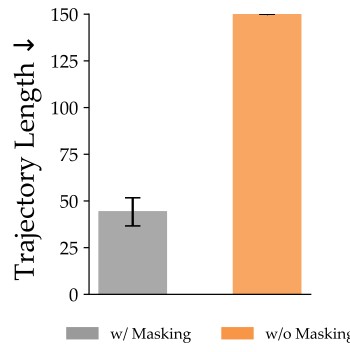

Figure 9: Ablation of the masking strategy for interpolated values.

## D  FULL TRAINING CURVES FOR ALL SIMULATION EXPERIMENTS

This section presents the complete training curves for all baselines across the simulation experiments in Figure 10. Unlike the main paper, which reports only the final converged values, these curves illustrate the learning dynamics and stability of each method throughout training.

*Note:* For BC and SSRR, which use different x-axes, we represent their results with a horizontal line indicating final performance.

## E  NETWORK ARCHITECTURES

**Actor and Critic Networks.** The actor and critic networks share the same architecture, differing only at the final layer: the actor outputs an action distribution, while the critic outputs a scalar value estimate. For MiniGrid-LfCD, we use a convolutional encoder with the following structure: $CONV(3, 2, 16)$ - ReLU - MaxPool(2,2) - $CONV(3, 2, 32)$ - ReLU - $CONV(3, 2, 64)$, followed by two fully connected layers of size 64. Here, $CONV(k, s, c)$ denotes a convolutional layer with kernel size $k$, stride $s$, and $c$ output channels.

For all other environments, we use separate 3-layer MLPs for the actor and critic, each with hidden layer size 256. For continuous control tasks, the final layer of the actor MLP outputs both the mean and standard deviation of a Gaussian distribution over actions. We use ReLU activations for MiniGrid-LfCD and tanh activations elsewhere.

**Goal Proximity Function and Discriminator.** The proximity function and discriminator networks adopt the same encoder architectures as the policy networks. For image-based inputs, we use the same convolutional encoder as above, followed by a single hidden layer of size 64. For other tasks, we use a 3-layer MLP with 64 hidden units. For uncertainty estimation, we maintain an ensemble of 5 proximity networks.

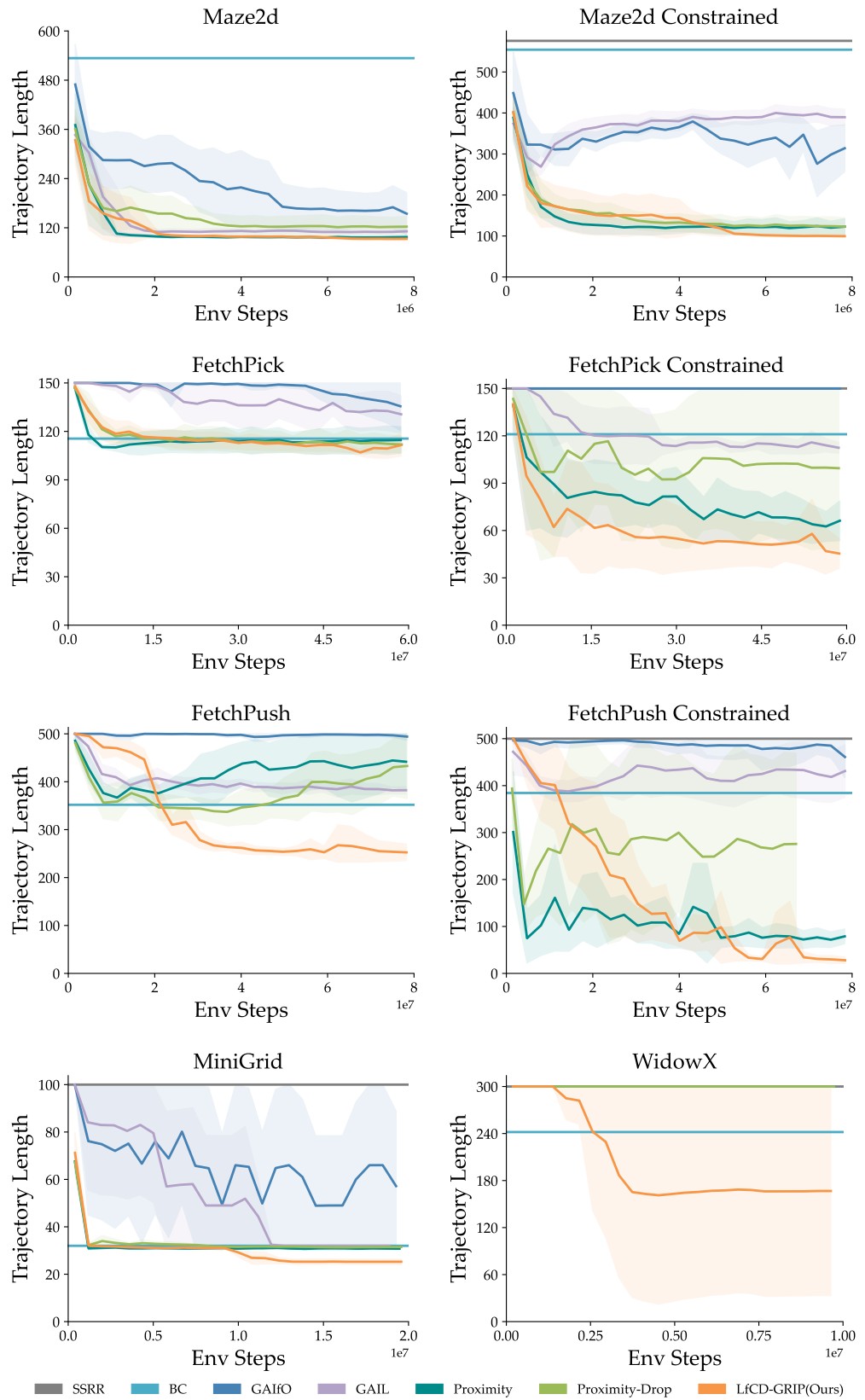

Figure 10: RL Training Curves: *UnconstrainedExpert* settings (left) and *ConstrainedExpert* settings (right) except Minigrid and WidowX. Both of them belong to *ConstrainedExpert* settings

## F  TRAINING DETAILS

For all baselines (except BC), we train policies using PPO (Schulman et al., 2017). A full list of training hyperparameters for each environment is provided in Table 2.

| Hyperparameter | Minigrid-LfCD | Maze2D | FetchPick | FetchPush | WidowX |
|---|---|---|---|---|---|
| **PPO related** | | | | | |
| Entropy Coefficient | 1e-2 | 1e-2 | 1e-3 | 1e-3 | 1e-4 |
| learning Rate | 1e-3 | 1e-3 | 1e-4 | 1e-4 | 1e-3 |
| Epochs per Update | 4 | 4 | 10 | 10 | 10 |
| Mini-batches | 4 | 4 | 32 | 32 | 32 |
| Rollout Size | 1e4 | 1e4 | 4096 | 4096 | 4096 |
| **Proximity Function related** | | | | | |
| Discount Factor $\delta$ | 0.95 | 0.95 | 0.99 | 0.99 | 0.95 |
| learning Rate $\delta$ | 1e-3 | 1e-3 | 1e-3 | 1e-4 | 1e-3 |
| Batch Size | 32 | 32 | 128 | 128 | 128 |
| Epochs for Pre-training | 2 | 5 | 2 | 5 | 500 |

Table 2: Policy-specific Hyperparameters

## G  DIFFERENCES FROM SUBOPTIMAL DEMONSTRATORS

Section 2 discusses learning from suboptimal demonstrations and explains that LfCD assumes constrained-but-competent demonstrators instead of noisy or inconsistent experts. Here, we elaborate the distinction more explicitly. Conceptually, suboptimal-demonstrator methods (e.g., T-REX, D-REX, SSRR) assume that expert and agent share the same action space, that the expert is unconstrained but imperfect, and that near-optimal behavior is already present in the data; their objective is to denoise or rank trajectories to recover the best actions hidden in suboptimal demonstrations. In contrast, LfCD assumes a competent expert who is structurally constrained by the interface (e.g., mode-switching joystick), while the robot later operates in a strictly larger action space. In this setting, the agent's optimal policy can lie beyond anything the expert can physically demonstrate, so the goal is to extract a goal-directed, state-only notion of progress from constrained trajectories and extrapolate efficiency beyond them. Table 3 summarizes these differences.

Table 3: Key differences between settings with suboptimal and constrained demonstrators.

| Property | Suboptimal demonstrators | Constrained demonstrators |
|---|---|---|
| Action space | Shared between expert and agent. | Expert acts in a restricted action subspace; agent has a larger space. |
| Expert behavior | Expert behaviors are noisy/inconsistent; some trajectories are better than others. | Expert is optimal under interface constraints (e.g., mode-switching joystick). |
| Agent policy | Assumes the agent's optimal policy is present in the expert data under noise. | Agent's optimal policy can lie beyond what the expert demonstrates |
| Problem objective | Denoise/rank trajectories to uncover the best actions in suboptimal data. | Extrapolate beyond demonstrated behavior to outperform the expert. |
| Learning signal | Relative quality of trajectories (rankings, noise levels). | Goal-reaching state-only progress to be independent of constraints. |
| Suitable application | Unconstrained but imperfect experts in a shared action space. | Interface- or embodiment-constrained experts with action-space mismatch. |

## H  TIME PENALTY DO NOT ENABLE REWARD EXTRAPOLATION

A natural alternative to our interpolation mechanism is to augment proximity-based IRL with a global time penalty, encouraging shorter trajectories. We analyze this modification and show that

temporal penalization does not address the fundamental limitation of reward extrapolation beyond expert coverage.

**Early Termination Bias.** In environments that terminate early upon failure, a time penalty encourages the agent to fail quickly in order to minimize cumulative penalty. In contrast, LfCD-GRIP continues to reward progress toward the goal and remains effective regardless of episode length.

**Uniform Penalization of Transitions.** A global time penalty treats all transitions equally, failing to distinguish between productive and unproductive steps. As a result, even sub-trajectories containing meaningful progress may be undervalued if embedded in longer episodes. LfCD-GRIP, by contrast, identifies and leverages high-quality sub-trajectories through interpolation, preserving valuable learning signals.

**Interaction with Dynamic Reward Learning.** Selecting an appropriate penalty scale is non-trivial. If the penalty is too large, it may overwhelm the learned reward signal. As proximity-based IRL uses a dynamic, evolving reward during training, this interaction becomes difficult to calibrate in practice.

**Empirical Evaluation.** Empirically, we ran an ablation on Maze2D comparing Proximity-IRL with various time penalty magnitudes. None matched the performance of LfCD-GRIP, further supporting the limitations of simple temporal penalization.

| Method | LfCD-GRIP | Proximity | +0.1 penalty | +0.01 penalty | +0.001 penalty |
|---|---|---|---|---|---|
| Avg. Episode Length | 101 | 112 | 119 | 111 | 112 |

Table 4: Maze2D-Constrained: Adding a global time penalty to Proximity-IRL does not recover the efficiency achieved by LfCD-GRIP.

## I  HYPERPARAMETER SENSITIVITY ANALYSIS

To assess the robustness of LfCD-GRIP, we conduct a hyperparameter sensitivity analysis across four key hyperparameters: (1) size of the expert dataset(12.5%, 25%, 50%, 100%), (2) number of pretraining iterations(2, 5, 10), (3) temporal decay factor $\delta$ used in proximity value calculation(0.9, 0.95, 0.99), and (4) the number of Monte Carlo samples $K$ used for confidence estimation(3, 5, 7, 10). The first experiment is conducted on MAZE2D, and the remaining three on MINIGRID.

For MINIGRID, the result reported in the main paper is 25.2 average episode length. The optimal path has a length of 24, while the constrained expert's trajectory length is 32. All baselines average above 32, meaning they fail to surpass the constrained expert. In contrast, LfCD-GRIP achieves over 25% improvement.

For MAZE2D, our reported result is an average trajectory length of 103, significantly outperforming the constrained expert whose average is 264.37. Also it obviously outperforms the best baseline, Proximity-IRL, with an average trajectory length of 113.

As shown in Table 5, except for pretrain iteration number = 10 and $K = 3$, LfCD-GRIP demonstrates strong and consistent performance across different hyperparameter choices.

Table 5: Hyperparameter Sensitivity Analysis

| Expert Dataset Size | Trajectory Length | Pretrain Iteration Number | Trajectory Length |
|---|---|---|---|
| 25 % | **104** | 2 | **25.2** |
| 50 % | **103** | 5 | **25.5** |
| 100 % | **105** | 10 | 33.5 |
| $\delta$ | Trajectory Length | $K$ | Trajectory Length |
| 0.9 | **25** | 3 | 29.5 |
| 0.95 | **25.2** | 5 | **25.2** |
| 0.99 | **25.0** | 7 | **25.3** |
| - | - | 10 | **24.1** |

