# OpenReview forum: "When a Robot is More Capable than a Human: Learning from Constrained Demonstrators"
_ICLR.cc/2026/Conference — ICLR 2026 Poster_

### Official Review · Reviewer_Q8E8 · 2025-10-23

**Soundness:** 2
**Presentation:** 3
**Contribution:** 2
**Rating:** 4
**Confidence:** 4

**Summary:**

This paper presents LfCD-GRIP, a new way to do imitation learning/learning from demonstration that considers the fact that the input expert demonstrations likely have some amount of "constraint" to them. The proposed approach seeks to learn better policies by using "unconstrained" actions. This is done by estimating the "confidence" in the "proximity" of different states to the goal, finding high-confidence states, and interpolating the proximity of the intermediate states. In this way a more generalized model of proximity can be used to guide the PPO RL loss function, which allows learning to visit states and take actions that may not have shown up in the expert demonstrations. The authors show results in simulation and on a real robot against several baselines, showing faster task performance, particularly in the setting where expert actions were constrained.

**Strengths:**

Originality
- The approach of identifying high-confidence good states and then interpolating between them is novel to me
Quality
- The paper compares against several baselines and has ablations
Clarity
* The graphs and visualizations are clear
* The paper is generally well-written
* The literature review is appropriately broad and deep for a problem like this
Significance
- All expert demonstrations are constrained in some way, so this general idea and the problem being tackled has broad applicability across imitation learning.

**Weaknesses:**

- The contribution of the paper, while novel, is fairly small. Proximity-based approaches are already published work (like the cited Lee et al. 2021). LfCD-GRIP only performs slightly better than the Proximity approach, with the small changes of estimating proximity confidence. Proximity interpolation is done by time, which is the same as Lee et al. 2021.
- There are so many figures in the paper that there is not much room for discussion of results.
* The results are not presented in a way that clearly shows that the proposed approach is better than prior work.
	* Graphs include error bars, but there is no indication of confidence levels or statistical significance.
	* No comparison to Ma et al. 2022, which was listed in the lit review and seems it seeks to solve almost the exact same problem.
- The explanation of the proposed approach is incomplete.
	* Line 251 says "we identify sub-trajectories where both endpoints are high-confidence, and use them as anchors for interpolation", but there is no description of how high-confidence endpoints are determined. I had to go to the code to determine how it is done, and it's not a fixed threshold but more complicated. This is an important part of the approach and should be described in detail.
* The experiments could more clearly illustrate the problem and proposed solution.
	* Most of the examples of a "constrained" expert are simply limiting the magnitude of the possible motion. This seems like an almost trivial constraint to overcome by using any amount of reward shaping during the RL process, like penalizing total time taken. However, I don't see evidence that this was tested against.
	* An experiment where the constraint was relaxed gradually (i.e. increasing the possible action interval in steps) to see that the proposed approach improves as the constraint is relaxed

**Questions:**

* Exactly how are "high-confident" states determined? I want to make sure I understood the code correctly.
* What is the intuition around how Proximity-Drop performs almost universally worse than Proximity? Did you test LfCD-GRIP without Drop - it seems like it might perform better?
* Conceptually, why wouldn't a technique like Proximity be able to learn to take diagonal actions in MiniGrid?

---

> ### Author Response · Authors · 2025-12-04
>
> We thank the reviewer for their thoughtful and constructive feedback. We are pleased with your comments on the novelty, clarity, and broad significance of our work. Below, we address each of your specific concerns and have made revisions based on your suggestions.
>
> ### W1. [Novelty] LfCD-GRIP proposes to (i) estimate confidence or (ii) interpolate proximity
>
> We respectfully disagree with the limited difference from Proximity-IRL due to the following reasons:
> - Conceptually, it is an incorrect claim that time-based interpolation is already part of Proximity-IRL, which instead assigns zero proximity values to all online (out-of-distribution) states and makes no attempt to estimate proximity for them. In contrast, LfCD-GRIP explicitly interpolates proximity values between confident states, enabling reward generalization and exploration beyond the states demonstrated.
> - Empirically, LfCD-GRIP significantly outperforms Proximity-IRL. For instance, in the MiniGrid experiment, only LfCD-GRIP is able to discover the diagonal shortcut, whereas all other baselines, including Proximity-IRL, fail to deviate from the constrained demonstration path. In the WidowX real-robot task, LfCD-GRIP and BC are the only methods that learn any meaningful policy; other baselines fail entirely. In other constrained environments, LfCD-GRIP improves success rates by at least 10% compared to the best baseline. Larger and clearer visualizations are provided in Appendix Figure 10.
>
> ### W2: “There are so many figures in the paper that there is not much room for discussion of results.”
>
> We believe the main paper provides considerable discussion of all the various experiments and analyses we performed, and we are not sure if there is anything specific that the reviewer wants to have further discussed. We will be happy to augment the discussion of results if the reviewer could kindly let us know what is missing.
>
> ### W3A: Error bars definition
> Error bars show ±1 standard deviation across 10 random seeds.
>
> ### W3B: Differences from VIP, Value-Implicit Pre-training (Ma et al. 2022).
> VIP (Ma et al., 2022) complements our approach of LfCD-GRIP, as these approaches address two orthogonal problems.
> - VIP is an offline goal-conditioned reinforcement learning objective to extract reward functions. It does not specify how the reward is used for downstream IRL.
> - LfCD-GRIP explicitly addresses how to enable the agent to assign reward functions to states not seen during reward learning, by (i) confidence filtering, and (ii) temporal interpolation from online data where the reward function is not confident.
> VIP’s strength lies in utilizing a large-scale dataset (4.3M frames) and large networks to solve vision-based goal-conditioned tasks, whereas LfCD-GRIP addresses the problem of learning from constrained demonstrations. Our approach is specifically designed to address constraint-induced distribution gaps, which VIP does not handle, and is therefore not a direct baseline.### W4 & Q1: Definition of high-confidence endpoint states
> At each training iteration, we compute the confidence threshold as the maximum variance of proximity predictions among expert states. Any online state with lower variance than this threshold is classified as high-confidence. This dynamic threshold ensures expert states are always included as anchors, while allowing confident online states to emerge progressively. Thanks to the reviewer comments, we have added further description of the confidence threshold in Section 4.2.

---

> ### Author Response · Authors · 2025-12-04
>
> ### W4 & Q1: Definition of high-confidence endpoint states
> At each training iteration, we compute the confidence threshold as the maximum variance of proximity predictions among expert states. Any online state with lower variance than this threshold is classified as high-confidence. This dynamic threshold ensures expert states are always included as anchors, while allowing confident online states to emerge progressively. Thanks to the reviewer comments, we have added further description of the confidence threshold in Section 4.2.
>
> ### W5. Why is the problem not as simple as adding a time penalty reward?
> We appreciate the reviewer’s suggestion of incorporating a time penalty into the base IRL reward. However, we believe this approach has several limitations:
> - **Early Termination Bias.** In environments that terminate early upon failure, a time penalty encourages the agent to fail quickly in order to minimize cumulative penalty. In contrast, LfCD-GRIP continues to reward progress toward the goal and remains effective regardless of episode length.
>
>
> - **Uniform Penalization of Transitions.** A global time penalty treats all transitions equally, failing to distinguish between productive and unproductive steps. As a result, even sub-trajectories containing meaningful progress may be undervalued if embedded in longer episodes. LfCD-GRIP, by contrast, identifies and leverages high-quality sub-trajectories through interpolation, preserving valuable learning signals.
>
>
> - **Tuning Difficulty.** Selecting an appropriate penalty scale is non-trivial. If the penalty is too large, it can overwhelm the learned reward signal. Since proximity-based IRL uses a dynamic, evolving reward during training, this interaction becomes difficult to calibrate in practice.
>
>
> Empirically, we ran an ablation on Maze2D comparing Proximity-IRL with various time penalty magnitudes. None matched the performance of LfCD-GRIP, further supporting the limitations of simple temporal penalization.
>
> | Baselines | LfCD-GRIP | Proximity-IRL + 0 penalty | Proximity + 0.1 penalty | Proximity + 0.01 penalty | Proximity + 0.01 penalty |
> |----|---|---|---|---|---|
> | Avg. Episode Length | **101** | 112 | 119 | 111 | 112 |
> ### Q2. Why Proximity-Drop performs worse than Proximity.
> Proximity-Drop adds dropout to all layers of the proximity network, which reduces network capacity and harms learning performance. However, dropout is necessary to estimate uncertainty: without it, the network produces deterministic outputs, making confidence estimation via variance impossible.
> ### Q3. Why Proximity-IRL fails to take diagonal actions in MiniGrid.
> In Proximity-IRL, all online states are assigned zero proximity, including those along efficient, goal-reaching trajectories not visited by the expert. As a result, transitioning from expert states to none-expert ones produces a large negative reward, discouraging the agent from exploring these regions. This leads to the agent strictly following expert-like paths and prevents discovery of more efficient solutions such as diagonal shortcuts.
>
> We appreciate the reviewer’s detailed comments and hope these clarifications address all concerns.

---

### Official Review · Reviewer_jvYJ · 2025-10-26

**Soundness:** 3
**Presentation:** 3
**Contribution:** 2
**Rating:** 2
**Confidence:** 4

**Summary:**

This paper proposes a framework for Learning from Constrained Demonstrations, Goal-proximity Reward InterPolation (LfCD-GRIP). LfCD-GRIP extends proximity-based IRL with confidence-guided reward propagation. The authors' formulation of LfCD is interesting, specifically focused on scenarios where the demonstrator is optimal within their constrained action space but where the robot has access to a broader set of actions. The methodology is well-designed and straightforward. The authors test their approach against several baselines and ablations of their proposed framework, and find that LfCD-GRIP has several benefits.

**Strengths:**

+ The paper has an interesting formulation of LfCD. I could see many applications where a robot may need to learn from LfCD.
+ The paper is well-written and clear.
+ The paper has ample results and a deployment on a real-world robot.

**Weaknesses:**

- The authors should better explain the difference between their framework and those that learn from suboptimal demonstrations. To my knowledge, works on learning from suboptimal demonstration often have similar motivation and can be applied to the same problem statement posed in this paper. This would also help better highlight the novelty in the proposed work.
- The results have very large standard deviations, and it is unclear whether LfCD-GRIP is actually outperforming other frameworks. A statistical significance analysis and explanation regarding large standard deviations would be beneficial.
- The key result noted in the intro (100 seconds to 12 seconds) seems an overclaim given the large standard deviation in Figure 8.

**Questions:**

1. Can the authors comment on why SSRR performs so poorly across these domains? From my knowledge of that framework, I cannot see a specific reason on why that framework should underperform GAIL by such a large margin.
2. How does the second term in Equation 2 help avoid overgeneralization?
3. Can you reply to the weaknesses noted above?

---

> ### Author Response · Authors · 2025-12-04
>
> We thank the reviewer for their thoughtful and constructive feedback. We are pleased that you appreciated the LfCD formulation’s impact, ample results, and real-robot experiment. Below, we address each of your specific concerns and have made revisions based on your suggestions.
> ### W1: Suboptimal v/s Constrained demonstrators: different assumptions and impact
> We appreciate and agree with the reviewer’s suggestion of better distinguishing constrained expert and suboptimal expert frameworks, in addition to what’s already covered in Section 2, since the difference is subtle but crucial.
>
> | Property | Suboptimal Demonstrators | Constrained Demonstrators |
> |----|---|---|
> | Action Space | Expert and agent share the **same action space**. | Expert acts in a **restricted action subspace**; agent has a **larger** space. |
> | Expert Behavior | Expert behaviors are **noisy / inconsistent**; some trajectories are better than others. | Expert is **competent under their interface constraints** (e.g., mode-switching joystick). |
> | Agent's Optimal Policy | Assumes the agent’s optimal policy is present in the expert data under noise. | The agent’s optimal policy can lie beyond what the expert can physically demonstrate**. |
> | Problem Objective | **denoise/rank trajectories** to uncover the best actions hidden in suboptimal data. | **extrapolate efficiency** beyond demonstrated behavior to outperform the expert. |
> | Learning Signal | **relative quality of trajectories** (rankings, noise levels). | **goal-reaching structure and state-only progress**, independent of constrained actions. |
> | Suitable Application | **Unconstrained but imperfect experts** in a shared action space. | **Interface- or embodiment-constrained experts** with action-space mismatch. |
>
> This comparison makes our distinction precise: works on suboptimal demonstrations assume that expert and agent share the same action space, that the expert is unconstrained but noisy, and that near-optimal behavior is already present in the data—so the objective is to denoise and rank trajectories to recover the best actions. In contrast, our LfCD setting assumes a competent expert who is structurally constrained by the interface, while the robot has a strictly larger action space; here, the optimal policy we seek can lie beyond anything the expert can physically demonstrate. As a result, LfCD focuses on extracting a goal-directed, state-only notion of progress from constrained demonstrations and extrapolating efficiency beyond them, rather than leveraging relative trajectory quality within a shared action space.
>
> We have added a full Section Appendix G, which includes the above table and distinction, to explicitly situate LfCD as complementary to suboptimal-demonstration methods rather than a special case. Thank you for helping us highlight the novelty more clearly.
> ### W2: Results are clearly statistically significant
> We agree that the results of some methods have large standard deviations; however, we emphasize that Figures 5, 8, and 10 clearly demonstrate that LfCD-GRIP outperforms other methods in constrained environments. The error bars provided already show that LfCD-GRIP outperforms other methods statistically significantly. The large standard deviations in performance are a direct result of RL on-policy training with PPO. The fact that LfCD-GRIP still outperforms the other methods beyond the standard deviation is a clear empirical strength.
> ### W3: Intro claim (100 seconds → 12 seconds) is on real robot, Figure 8 is a simulation.
> We emphasize that the intro claim (100 seconds → 12 seconds) is correct, because it is the result of deploying the successful seeds for BC and LfCD-GRIP on the real robot. We found that BC’s average episode length was 10× longer than LfCD-GRIP. Figure 8 results are based on the simulation experiment, which still demonstrate a similar level of improvement due to LfCD-GRIP.

---

> ### Author Response · Authors · 2025-12-04
>
> ### Q1: Why SSRR performs poorly?
>
> The primary reason SSRR underperforms compared to GAIL is that SSRR can generate misleading ranking labels, resulting in poor reward models. This behavior has been reported in prior work, such as CAIL (Confidence-Aware Imitation Learning from Demonstrations with Varying Optimality) [1], where SSRR failed to learn useful rewards in Reacher, Ant, and Franka Panda Arm tasks, while GAIL performed reasonably well. We agree with their analysis: SSRR assumes that noise-injected demonstrations correlate inversely with performance, which means the larger the noise, the worse the performance. However, this is not always true and, thus, SSRR can generate misleading ranking labels, leading to poor reward models. GAIL, on the other hand, only matches expert behavior and does not rely on such assumptions, making it more stable in practice.
> **[1] Zhang, Songyuan, et al. "Confidence-aware imitation learning from demonstrations with varying optimality." Advances in Neural Information Processing Systems 34 (2021): 12340-12350.**
> ### Q2: How does the second term in Equation 2 help avoid overgeneralization?
>
> The second term in Eq. 2 corresponds to the regularization loss in Proximity IRL. As discussed in Section 3.2 of their paper, training on expert demonstrations alone can lead to overfitting. To prevent this, they assign zero proximity to states from the agent’s own trajectories, to discourage any exploration of states not visited by the expert. This is analogous to the negative sampling approach in adversarial imitation learning. As shown in Figure 8 of their appendix, omitting this regularizer results in total failure to learn a useful reward function. In contrast, our method does not block the agent from deviating from expert states, because that’s where it can find more efficient paths. Therefore, we split the second term from Eq. 2 into (i) states whose proximity values can be interpolated confidently, and (ii) states whose proximity values should be kept zero.
>
> We thank the reviewer again for their valuable feedback and hope this response sufficiently clarifies our contributions and experimental results.

---

### Official Review · Reviewer_pvLM · 2025-10-27

**Soundness:** 2
**Presentation:** 3
**Contribution:** 3
**Rating:** 6
**Confidence:** 2

**Summary:**

The paper focuses on the setting where the demonstrator performance is hindered due to constraints in observations/actions.
The paper questions whether it is possible to create a policy, without said constraints, to outperform the constrained demonstrator using the generated demonstrations.
The paper proposes to learn a goal-proximity reward with a confidence-based proximity interpolator, that consists of a confidence estimator and a trajectory-wise interpolator.
The former estimates the confidence through Monte-Carlo dropout and aims to identify reliable observations, and the latter estimates low-confidence observations through interpolating the values between reliable observations.
The paper then conduct experiments on four simulated tasks and a real-world manipulation picking task, and demonstrate that the proposed method identifies out-of-constraint actions that result in shorter trajectories.

**Strengths:**

- The introduced problem is well motivated, especially in the robotic setting where the demonstrator may have less degree-of-freedoms compared to the actualy embodiment.
- The proposed idea is straightforward and easy-to-follow, and the writing is clear.
- The analyses on MiniGrid environment and out-of-constraint actions are great to demonstrate that the proposed method identifies better actions beyond the demonstrator.

**Weaknesses:**

I am happy to increase my score after these comments are addressed.
- While the confidence estimation module will provide low variance on "reliable observations", the uncertainty might come from another source, e.g., having multiple demonstrators gathering data.
	- As we scale the number of demonstrators, this might become a degenerate problem where none of the demonstrations contain high-confidence states.
- There are a handful of design/hyperparameter choices that I am unsure if it's well justified/experimented.
	- Decaying strategy for masking
	- What's considered as high confidence?
	- The choice of interpolation strategy
	- There are no sensitivity analysis on any hyperparameters, especially on the introduced modules
		- $\delta$, $K$, etc.
- Nit: It would be great if the paper shows the expert performance on all figures.

**Questions:**

- About Eq. 5: Is log-scale goal proximity distances the same as $log(r_{end}) - log(r_{start})$?

---

> ### Author Response · Authors · 2025-12-04
>
> Thank you for the insightful feedback. We are glad that the motivation is strong and the approach is clear. Our responses to the raised concerns and the revisions made are provided below.
> ### W1. Robustness to different sources of uncertainty.
>  You raise an important point regarding the impact of demonstrator variation. In our current setup, we assume all demonstrators operate under the same constraints (e.g., using the same interface). As a result, the proximity values remain consistent for shared states, since temporal distance to the goal is invariant across demonstrations under the same constraint. Thus, expert states maintain low uncertainty. Extending our method to handle heterogeneous demonstrators with different constraints is an interesting future direction.
> ### W2. Hyperparameter design and sensitivity.
> We briefly explain the design motivation behind key components:
>
> - **Decay strategy:** The decay value starts at 100% and linearly anneals to 0 at the end of training. This encourages initial behavior to align with standard proximity IRL while gradually enabling generalization beyond expert coverage.
>
>
> - **Confidence threshold:** At each iteration, the threshold is dynamically set as the maximum proximity variance among expert states. States with lower variance than this threshold are treated as high-confidence. This ensures that expert states are always classified as high-confidence anchors during training. The confidence threshold description is updated in Section 4.2.
>
>
> - **Interpolation strategy:** The interpolation follows the exponential decay structure of proximity values. Because expert proximity reflects decayed temporal distance, interpolated proximity values along agent trajectories are also computed via exponential decay.
>
>
> We also conducted sensitivity analysis on the two suggested hyperparameters:
> $\delta$ (0.9, 0.95, 0.99)
> $K$ (3, 5, 7, 10)
> | $\delta$ | Avg. Trajectory Length | $K$ | Avg. Trajectory Length |
> |---|---|---|---|
> |0.9 | **25.0** | 3 | 29.5 |
> |0.95 | **25.2** | 5 | **25.2** |
> |0.99 | **25.0** | 7 | **25.3** |
> |- | - | 10 | **24.1** |
>
>
> Except for $K$ = 3, the results are robustly stable. Results are presented in Appendix H, showing that LfCD-GRIP remains stable across different hyperparameter values, indicating LfCD-GRIP’s robustness.
> ### W3. Expert performance in figures.
>  Thank you for this suggestion—we have updated our figures to include expert performance for better comparison.
> ### Q1. Clarification on log-scale proximity (Section 4.3).
>  Yes, your understanding is correct: $\rho_\text{start}$ and $\rho_\text{end}$ represent the log-transformed proximity values of the start and end states, i.e., $\rho = \log(\text{proximity})$.
>
> We appreciate your thoughtful comments and hope this addresses your concerns.

---

### Official Review · Reviewer_25Mi · 2025-11-01

**Soundness:** 3
**Presentation:** 3
**Contribution:** 3
**Rating:** 4
**Confidence:** 3

**Summary:**

This paper proposes LfCD, a learning from demonstration method that specializes scenarios where the demonstrators are "constrained" compared to the robot. LfCD extends the IRL framework by 1) using a state-only, learned proximity reward and 2) a mechanism to enhance fidelity of the proximity reward on OOD observations. The core idea of 2) is to provides reward target computed from linearly interpolating confident states to the less confident ones.

**Strengths:**

+ The paper is well written and presented. The algobox provides a clear overview of the presented approach.
+ The core idea of creating pseudo supervision using confident predictions makes a lot of sense

**Weaknesses:**

- It seems the presented approach only works in setups where the state space is 1) observable / low-dimensional and 2) smooth. If 1) or 2) does not hold, linear interpolation might not be a proper way to obtain the supervision targets.
- While the presented approach makes sense, I don't quite see how it relates to the motivation of expert being constrained. The proposed method seems rather generic.
- Similarly, I would like to see how the presented approach improves over the baseline, over various levels of the demonstrator being "constrained".
- I'd like to see a more detailed analysis on the sensitivity of the hyperparameters of the method, such as the confidence/non-confidence threshold, data / compute of the pretraining of the network vs training etc.

**Questions:**

Similar to the weakness section above.

- Can the presented approach apply to non-smooth, or high-dimensional state space, or how can one extend the method to make it work on them?
- How does the presented approach improve over the baseline over various "constrained" level of the demonstrator?
- Can we get a more detailed analysis on the sensitivity of the hyperparameters of the method, such as the confidence/non-confidence threshold, data / compute of the pretraining of the network vs training etc.

---

> ### Author Response · Authors · 2025-12-04
>
> We thank the reviewer for their time and constructive feedback. We are pleased that you found the paper clear and agree with the idea of pseudo self-supervision labels. Below, we address each of your specific concerns, and have made revisions based on your suggestions.
>
> ### W1. Interpolation is in temporal space, not state space
> > It seems the presented approach only works in setups where the state space is 1) observable / low-dimensional and 2) smooth. If 1) or 2) does not hold, linear interpolation might not be a proper way to obtain the supervision targets.
>
> We clarify a misunderstanding here: **LfCD-GRIP does not make any state space assumptions** regarding observability or smoothness. The “linear” interpolation described in Section 4.3 is a **temporal interpolation** (see Eq. 5) to assign goal-proximity supervision targets to low-confidence observations. The observations themselves can take any form, as already evidenced in our experiments: **high-dimensional**, such as the **image-based** 19 x 19 x 4 observations in MiniGrid (Section 5.1), or **non-smooth**, such as the **dexterous** manipulation tasks of FetchPick, FetchPush, and WidowX.
>
> ### W2. Our approach is indeed useful beyond constrained experts
> We fully agree that the proposed insight in this work has benefits beyond those of constrained experts. Concretely, our proposed approach of *interpolation of proximity rewards across an agent’s trajectories* allows LfCD-GRIP to generalize its reward function to unseen states, which in turn allows the agent to explore widely and find more efficient solutions. Thus, our key insight is useful for any problems where the state coverage of the expert dataset is limited, and more efficient paths to the goals exist. Since the problem of constrained experts is the most practically important application for this insight, we demonstrate the major benefits of LfCD-GRIP in this setting and plan to explore broader applications of proximity interpolation in future work.
> Based on your comment, we have added a 'Future Work' section to enhance our paper’s message.
>
> ### W3. Varying constraint-level of expert already in Section 5.4
> The reviewer asked to see how our approach improves over baselines for different levels of expert constraints. We highlight Section 5.4, which does exactly that: LfCD-GRIP maintains strong performance across both constraint levels, whereas baselines such as BC and Proximity-based IRL degrade substantially under the more severe constraints.
>
> ### W4. Added Further Analysis: Sensitivity to dataset size and training budget
> We have added hyperparameter sensitivity analysis, three seeds each, on:
> - Dataset size (25\%, 50\%, 100\%)
> - Pretraining Budget, i.e. pretraining iteration numbers (2, 5, 10)
> - **Confidence threshold is auto-tuned**: We emphasize that the confidence threshold is dynamically set to the maximum confidence value among expert states in each iteration. This ensures that expert states are always classified as high-confidence anchors during training. The confidence threshold description is updated in Section 4.2.
>
> | Expert Dataset Size | Avg. Trajectory Length | Pretrain Iteration Number | Avg. Trajectory Length |
> |---|---|---|---|
> |25 \% | 103 | 2 | 25.2 |
> |50 \% | 103 | 5 | 25.5 |
> |100 \% | 107 | 10 | 33.5 |
>
>
> According to the experiment, except for pretraining iteration numbers=10, all experiment results are robustly stable. We also attached the detailed experiment results in Appendix H, indicating LfCD-GRIP’s robustness.
>
> We thank the reviewer again for their valuable suggestions and hope that our response addresses all the concerns adequately, supported by both new and existing experiments.

---

### Author Response · Authors · 2025-12-04

We sincerely thank all reviewers for their valuable and constructive feedback. Reviewers consistently highlighted the clarity of the paper, the novelty of temporal interpolation as a mechanism for reward generalization, and the significance of studying constrained experts. In response, we made substantial revisions and added new analyses that collectively strengthen our contributions. Below we summarize the major updates.

----
# Summary of revisions and clarifications
----
## 1. Added distinction description between suboptimal vs. constrained experts (Reviewer jvYJ)
- We added a full Appendix G comparing suboptimal-demonstration work vs. our constrained-expert setting (with a table).


- We highlight three key distinctions:


  - Action-space mismatch (expert restricted; agent larger).


  - Expert is **competent under their interface constraints**.


  - LfCD’s goal is to **extrapolate efficiency** beyond demonstrated behavior to outperform the expert, but not **denoise/rank trajectories** to uncover the best actions hidden in suboptimal data.


- This clarifies that LfCD is not a special case of suboptimal-demo methods but addresses a fundamentally different challenge.

----
## 2. Improved core algorithm description: dynamic confidence threshold + interpolation in log-space (Reviewer pvLM, Q8E8)
- **Dynamic confidence threshold:**
 Now clearly stated in Section 4.2: threshold = max proximity variance over expert states, which is recalculated in every iteration, ensuring experts are always anchors while allowing confident online states to emerge.


- **Interpolation in log-space:**
 Section 4.3 now explains that interpolation operates on log-transformed proximities to match the exponential decay structure of temporal distance.


----
## 3. Interpolation is temporal, not spatial; no assumptions on state smoothness (Reviewer 25Mi)
- We clarified that LfCD-GRIP performs **temporal interpolation** over agent trajectories (Eq. 5), **not spatial or geometric interpolation**, so the method imposes no requirements on state-space smoothness or observability.


- We emphasize that LfCD-GRIP already handles high-dimensional, non-smooth observations, validated in MiniGrid (image-based, 19×19×4), FetchPick/Push, and WidowX (dexterous manipulation).
----
## 4. Hyperparameter Robustness and Clarified Baseline Behaviors (Reviewers 25Mi, pvLM, jvYJ)
 We incorporated several reviewer-requested updates for experiments:
- Expert performance baselines added to all relevant figures for easier comparison.
- Extended sensitivity analysis added to Appendix H. As the following table shows, LfCD-GRIP is robustly stable.
| Expert Dataset Size | Avg. Trajectory Length | Pretrain Iteration Number | Avg. Trajectory Length | $\delta$ | Avg. Trajectory Length | $K$ | Avg. Trajectory Length |
|---|---|---|---|---|---|---|---|
|25 \% | **103** | 2 | **25.2** |0.9 | **25.0** | 3 | 29.5 |
|50 \% | **103** | 5 | **25.5** |0.95 | **25.2** | 5 | **25.2** |
|100 \% | **107** | 10 | 33.5 |0.99 | **25.0** | 7 | **25.3** |
| - | - | - | - | - | - | 10 | **24.1** |



- Added explanation of why Proximity-IRL + time-penalty cannot replace our method (Maze2D ablation included).
- Clarified SSRR’s poor performance using prior findings (Zhang et al., NeurIPS 2021).
These additions make empirical comparisons clearer and strengthen claims of robustness.
----
## 5. New “Future Work” section + clarifications of broader applicability (Reviewer 25Mi, pvLM)
- Added Future Work section discussing broader applicability beyond constrained experts, such as limited-coverage expert datasets and heterogeneous demonstrators.
- Clarified that the key insight—interpolating proximity across trajectories to generalize rewards beyond demonstrated states—is broadly useful wherever expert coverage is limited.

---

### Meta-Review · Area_Chair_JtD7 · 2025-12-23

**Summary:**

The authors investigate a novel variation of learning from demonstrations for which they propose a simple and well-motivated approach. The reviewers questioned the needed assumptions (e.g., wrt the state space or diversity of demonstrators), the genericity of the proposition vs its motivation, the motivation behind the design of some components, the experimental evaluation (e.g., design of evaluation set-up, sensitivity wrt to how constrained a demonstrator is or wrt other hyperparameters, interpretation of empirical results), the differentiation wrt work on learning from suboptimal demonstrations, the significance of the contribution, or lack of details in the description of the method.

**Reviewer Concerns:**

In their rebuttal, the authors clarified missing or unclear points in their paper (e.g., high-confidence endpoints or relation with previous methods), corrected misunderstanding in the reviews (e.g., temporal vs spatial interpolation), provided additional experimental results (e.g., sensitivity). The main point that I believe the authors could have answered better corresponds to the case where there are several demonstrators. It seems that the authors implicitly preclude the multimodal case in the demonstrations where completely different trajectories are acceptable. This case can actually also occur even when there is one expert demonstrator.

**Reviewer Scores:**

The two reviewers (25Mi and jvYJ), who have the most negative scores, seem to have based their evaluation on some misunderstandings. The former reviewer incorrectly believed that the method was based on spatial interpolation, questioned about the genericity of the proposition vs its original motivation, and asked for further experimental results (sensitivity). The latter mainly questioned two points: (1) the difference between the proposed variant of LfD and the existing one based on suboptimal demonstrations; (2) the validity of the empirical conclusions. In their rebuttal, the authors address all those points in a satisfying way. I believe that those two reviewers would have raised their scores.

Given the rebuttal, I don't think that the other reviewers would have decreased their scores.

---

### Decision · Program_Chairs · 2026-01-26

Accept (Poster)